# Designing diverse and high-performance proteins with a large language model in the loop

**Carlos A. Gomez-Uribe** *, **Japheth Gado, Meiirbek Islamov**

Solugen, Inc., Houston, Texas, United States of America

* cgomez@alum.mit.edu

## Abstract

We present a protein engineering approach to directed evolution with machine learning that integrates a new semi-supervised neural network fitness prediction model, Seq2Fitness, and an innovative optimization algorithm, **b**iphasic **a**nnealing for **d**iverse and **a**daptive **s**equence **s**ampling (BADASS) to design sequences. Seq2Fitness leverages protein language models to predict fitness landscapes, combining evolutionary data with experimental labels, while BADASS efficiently explores these landscapes by dynamically adjusting temperature and mutation energies to prevent premature convergence and to generate diverse high-fitness sequences. Compared to alternative models, Seq2Fitness improves Spearman correlation with experimental fitness measurements, increasing from 0.34 to 0.55 on sequences containing mutations at positions entirely not seen during training. BADASS requires less memory and computation compared to gradient-based Markov Chain Monte Carlo methods, while generating more high-fitness and diverse sequences across two protein families. For both families, 100% of the top 10,000 sequences identified by BADASS exceed the wildtype in predicted fitness, whereas competing methods range from 3% to 99%, often producing far fewer than 10,000 sequences. BADASS also finds higher-fitness sequences at every cutoff (top 1, 100, and 10,000). Additionally, we provide a theoretical framework explaining BADASS's underlying mechanism and behavior. While we focus on amino acid sequences, BADASS may generalize to other sequence spaces, such as DNA and RNA.

## Author summary

Designing proteins with enhanced properties is essential for many applications, from industrial enzymes to therapeutic molecules. However, traditional protein engineering methods often fail to explore the vast sequence space effectively, partly due to the rarity of high-fitness sequences. In this work, we introduce BADASS, an optimization

**Data availability statement:** The code and data for this work are available at https://github.com/SoluLearn/BADASS.

**Funding:** The author(s) received no specific funding for this work.

**Competing interests:** The authors have declared that no competing interests exist.

algorithm that samples sequences from a probability distribution with mutation energies and a temperature parameter that are updated dynamically, alternating between cooling and heating phases, to discover high-fitness proteins while maintaining sequence diversity. This stands in contrast to traditional approaches like simulated annealing, which often converge on fewer and lower fitness solutions, and gradient-based Markov Chain Monte Carlo (MCMC), also converging on lower fitness solutions and at a significantly higher computational and memory cost. Our approach requires only forward model evaluations and no gradient computations, enabling the rapid design of high-performing proteins that can be validated in the lab, especially when combined with our Seq2Fitness models. BADASS represents a significant advancement in computational protein engineering, opening new possibilities for diverse applications. Our code is publicly available at https://github.com/SoluLearn/BADASS.

## Introduction

Protein engineering plays a crucial role in biotechnology due to the transformative potential of high-performance proteins across a wide range of applications. Traditional approaches, such as directed evolution, are often time-consuming and labor-intensive, prone to becoming trapped in local optima, and limited to sequences with mostly a single mutation away from the starting sequence in each screening iteration [1]. In protein design, fitness is a quantitative description of the desired protein function, such as the conversion of substrate into product in a chemical reaction catalyzed by an enzyme. Recently, machine learning has been demonstrated to accelerate the discovery of proteins with improved fitness by overcoming limitations faced by traditional directed evolution through accurate prediction of fitness for sequences with multiple mutations, facilitating the in-silico exploration of broader regions of the sequence space [2–5].

Effective protein design with machine learning generally involves two key steps: first, building an accurate predictive model of protein fitness, and second, using this model to design a library of protein sequences that optimize the predicted fitness [6]. In recent years, protein language models have emerged as the state-of-the-art approach for predicting the effects of mutations on protein fitness [7,8]. However, zero-shot application of these models infers fitness from the distribution of amino acids in evolutionary data, which may diverge from experimentally measured or phenotypical fitness, particularly when the phenotypical fitness was not a target of evolutionary selection—a common occurrence in biotechnological applications [9]. Semi-supervised learning, integrating evolutionary predictions from zero-shot inference with experiment labels, has been shown to produce models that significantly improve the accuracy of phenotypical fitness prediction [10,11].

Even with a fairly accurate model for predicting fitness, protein design efforts with machine learning may fail in the second step, if they do not produce high-fitness sequences that can be validated in the lab [6]. Generating a diverse set of high-fitness sequences in-silico maximizes the probability of finding proteins in the lab with the desired function and properties, such as stability or high expression levels. Since the sequence fitness landscape is discrete and vast, and predictive models are typically large and computationally expensive to evaluate, efficiently identifying high-scoring sequences, which are rare within the rugged fitness landscape, is often a significant challenge [12,13]. Several approaches have been proposed to address this challenge, including samplers based on diffusion models [14,15], and gradient-based methods like EvoProtGrad and GGS [16,17]. However, common optimization techniques often struggle to efficiently navigate the vast sequence space [18,19]. These challenges

arise partly due to their intensive computational requirements, which limit the number of sequences explored under a fixed computational budget. As a result, the field is increasingly focusing on developing methods to identify diverse, high-fitness protein sequences using fitness models integrated within the optimization loop [16,17,20].

This paper presents a new approach for directed evolution with machine learning (Fig 1). Our method integrates semi-supervised neural networks, named Seq2Fitness, which leverage protein language models to infer the fitness landscape from evolutionary density and experiment data. We also propose a novel protein sequence optimization algorithm–biphasic annealing for diverse adaptive sequence sampling (BADASS)–to design high-performance proteins with the Seq2Fitness or any other sequence-to-fitness machine learning models, requiring relatively few evaluations of the model without need for computing gradients. We compare our approach with the current alternatives described in [16] (EvoProtGrad) and [17] (GGS), demonstrating superior performance across design tasks using alpha-amylase (AMY_BACSU) [21] and an endonuclease (NucB) [3]. We also developed a theory to motivate BADASS and explore why it works.

## Results

### Predicting protein fitness with Seq2Fitness

We developed a model, Seq2Fitness, to predict protein fitness from sequence. Seq2Fitness utilizes embeddings, log probabilities, and zero-shot scores from the ESM2-650M language model, and zero-shot scores from the ESM2-3B language model [22]. It employs parallel convolutional paths with novel statistical pooling layers to map sequence variants to experimental fitness measurements; see Fig 1 and Materials and Methods section for more detail.

To evaluate Seq2Fitness, we selected four fitness datasets representative of real-world protein fitness applications [23]: GB1 (a binding protein)[24], AAV (a viral protein)[4], NucB[3],

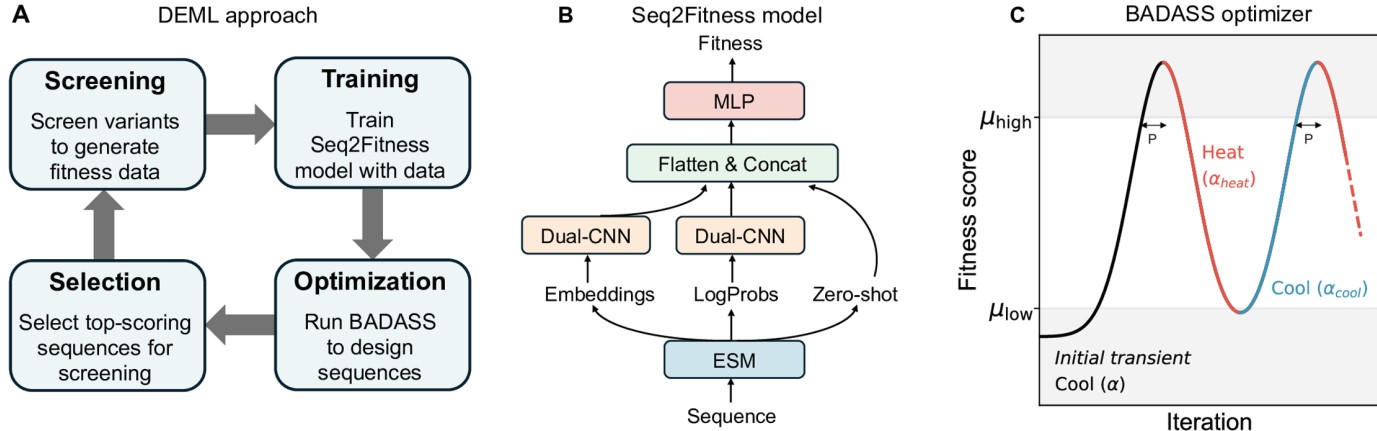

**Fig 1. Overview of our approach**. (A) Pipeline for directed evolution with machine learning using our proposed fitness prediction model and optimizer. New designs can start by running BADASS to select sequences with multiple mutations and high zero-shot scores from evolutionary models like ESM2 for an initial screening. (B) Architecture of the semi-supervised Seq2Fitness model. (C) The optimizer approach demonstrating the initial transient phase and two cooling and heating cycles. Sequences are iteratively sampled from an updated probability distribution, with the sampling temperature reduced (cooled) as the average fitness score rises. After the set point $\mu_{\text{high}}$ is breached, cooling continues for *patience* number of iterations, after which the temperature is increased (heated). The fitness score decreases until it reaches the set point $\mu_{\text{low}}$, at which point cooling resumes. The optimization process consists of multiple cooling and heating phases.

and AMY_BACSU [21](two enzyme proteins). Notably, these datasets are rich in multi-mutant variants, extending beyond single-point mutations, making them ideal for evaluating the ability of models to capture epistasis and predict higher order mutations. Using these datasets, we benchmarked Seq2Fitness against selected state-of-the-art methods from different model types to ensure a comprehensive comparison across model types while minimizing redundancy and computational cost. For zero-shot approaches [8,25–27], we compared with fitness scores from the ESM2-650M model [22]. For supervised methods [28], we compared with the CNN approach from Gelman et al. [9,18]. For semi-supervised methods [29], we compared with the augmented model from Hsu et al [10].

We evaluated the models with different training/test dataset splits that collectively assess the ability of the model to extrapolate to new sequences, to new regions of the sequence space with a higher number of mutations, and to novel mutations beyond those seen in the training data. The dataset splits we used include (i) an 80/20 random sequence split, (ii) a two-vs-rest split [23], where all sequence variants with up to two mutations were included in the training set and the remainder in the testing set, and (iii) a mutational and (iv) a positional split [9,30], where mutations or mutated positions in the test set were not present in any sequences in the training set. We used 80/20 train-test splits for computational efficiency, although 5-fold cross-validation could provide more robust performance estimates with sufficient computational resources. Our focus on challenging extrapolation tests (mutational, positional, and two-vs-rest splits) provides stringent evaluation of real-world generalization capability, which is the primary concern in protein engineering applications.

Our results showed that Seq2Fitness consistently outperformed the other models across dataset splits with superior average scores across the AAV, GB1, NucB and AMY_BACSU datasets (Table 1). The results on individual data splits showed that the improvement in performance of Seq2Fitness was most pronounced when extrapolating to new mutations and positions not present in the training set. Specifically, Seq2Fitness achieved average scores of 0.72 and 0.55 on mutational and positional splits, respectively, compared to 0.59 and 0.34 for the next best models, representing a 22% and 64% improvement in scores, respectively. The results on individual datasets and splits are found in Tables A–D in S1 Text. Additionally, we found that removing components of the Seq2Fitness architecture led to decline in performance, on average, highlighting the importance of each component to overall performance. These results, in Tables E–H in S1 Text, underscore the ability of Seq2Fitness to learn the fitness landscape effectively, enabling it to make accurate generalizations for designing new sequences.

**Table 1. Performance comparison of Seq2Fitness and alternative models**. Models were evaluated across different train/test splits, with performance metrics evaluated as Spearman correlation for regression tasks and adjusted AUC for classification tasks (NucB). Scores represent averages across the AAV, GB1, NucB, and AMY_BACSU datasets.

| Model | Random | 2-vs-rest | Mutational | Positional | Average |
|---|---|---|---|---|---|
| Zero-shot | 0.27 | 0.31 | 0.13 | 0.34 | 0.26 |
| Linear one-hot | 0.74 | 0.54 | 0.53 | -0.10 | 0.43 |
| Linear ESM | 0.80 | 0.55 | 0.35 | 0.26 | 0.49 |
| Aug. ESM | 0.75 | 0.57 | 0.47 | 0.31 | 0.53 |
| CNN one-hot | 0.87 | 0.48 | 0.45 | 0.08 | 0.47 |
| CNN AAindex | 0.86 | 0.46 | 0.40 | -0.05 | 0.42 |
| CNN ESM | 0.78 | 0.39 | 0.59 | 0.23 | 0.50 |
| Seq2Fitness | **0.88** | **0.66** | **0.72** | **0.55** | **0.70** |

## BADASS: Biphasic Annealing for Diverse Adaptive Sequence Sampling

We designed BADASS to efficiently explore the vast sequence landscape and identify diverse high-fitness variants. BADASS operates by iteratively sampling and scoring batches of sequences with a provided fitness model, such as Seq2Fitness. Each batch of sequences is sampled from a probability distribution that is dynamically updated based on mutation energies computed from the scores of previously evaluated batches, and a temperature parameter that is adjusted as the optimization progresses. In contrast to traditional simulated annealing [31,32]–which gradually cools the system but leaves the energy function constant, often resulting in fewer high-scoring sequences and reduced diversity, due to a rapid decline in fitness score variance (Fig A in S1 Text)–BADASS utilizes dynamic temperature control to sustain a high score variance throughout the optimization process. As the optimization progresses, BADASS typically results in oscillations between regions with low and high scores across iterations. The mutation energies are also updated at every iteration, resulting in a dynamic approach that prevents premature convergence and promotes the discovery of more diverse sequences with higher scores. We developed theory to explain why the particular combination of dynamic temperature and mutation energy adjustments, along with the specific form of the mutation energies, makes BADASS an effective optimization approach to explore sequence space.

## Performance of BADASS in protein optimization

We evaluated BADASS on protein design tasks, specifically to identify higher-scoring alpha-amylase (AMY_BACSU) [21] and endonuclease (NucB) [3] sequences, with both zero-shot scores from ESM2-650M or Seq2Fitness predictions as the fitness score. Fig 2 shows the average sequence score and its variability when exploring sequence space for the alpha-amylase tasks with BADASS. For each task, BADASS was benchmarked against EvoProtGrad [16] and GGS [17], two recent gradient-based MCMC approaches. Both methods use proposal distributions that depend on gradients of the fitness score with respect to input amino acids. While this gradient information aims to guide sampling, it comes with two significant drawbacks. First, computing these gradients is computationally intensive, making these methods significantly slower than BADASS which requires only forward model evaluations. Second, as our experiments demonstrate, the gradient-based sampling appears to limit exploration–both methods find fewer high-fitness sequences than BADASS, with EvoProtGrad particularly struggling to achieve good coverage of sequence space in tasks like NucB optimization. GGS attempts to address this limitation through dataset augmentation and smoothing, but our results show this intensive approach does not clearly improve performance over the original fitness landscape. EvoProtGrad did not originally incorporate a temperature parameter for the case of a single model, in effect defaulting to a temperature of 1.0. To improve EvoProtGrad's performance, we also tested it with a temperature of 0.1, equivalent to the temperature value used in GGS for the otherwise identical MCMC sampling distribution. We also modified the code to ensure consistency across sequence scores obtained during the EvoProtGrad process and during re-evaluation with the same scoring model.

Tables 2 summarizes the alpha-amylase optimization results, demonstrating that BADASS consistently outperformed EvoProtGrad in finding sequences with superior fitness scores. Specifically, 100% of the top 10,000 sequences generated by BADASS consistently achieved higher fitness scores than the reference sequence using both ESM2 and Seq2Fitness across sequence subspaces with different numbers of mutations $k$. In contrast, as few as 3.52% to 42.6% of the top 10,000 sequences found by EvoProtGrad are better than the reference sequence across temperatures and number of mutations for Seq2Fitness, and 90.1% to 99.5%

**Table 2. Alpha amylase sampling:** Performance comparison between BADASS, EvoProtGrad and GGS (using EvoProtGrad on the Smoothed Seq2Fitness model) using ESM2 and Seq2Fitness models. All approaches are given comparable GPU compute time for the sampling. GGS requires an additional round to evaluate sequences with the original Seq2Fitness model. Metrics include the percentage of sequences better than wild type in the top 10,000 sequences found (or less when a method cannot find enough), the best, best 100th, and best 1,000th sequence scores, and the number of unique mutations and unique mutated sites present in the top 10,000 sequences. The number of mutations per sequence is $k$. As benchmarks, the reference alpha amylase sequence has an ESM2 score of 0.0, and a Seq2Fitness score of 0.8. BADASS was run for 200 iterations with a batch size of 520 sequences. Missing entries for EvoProtGrad (using T=0.1 for ESM2) are due to the generation of a limited number of unique sequences (on the order of hundreds), as the sampler becomes overly concentrated on a small number of mutations.

| | | BADASS | | | EvoProtGrad | | | |
| --- | --- | --- | --- | --- | --- | --- | --- | --- |
| | | Gamma = 1 | | | T = 0.1 | | T = 1.0 | |
| Fitness | Metric | k = 2-6 | 2 | 6 | 1-2 | 1-6 | 1-2 | 1-6 |
| **ESM2** | % better than WT | **100** | **100** | **100** | 97.67 | 99.52 | 90.10 | 97.00 |
| | Best score | **55.91** | 20.19 | 54.12 | 18.80 | 54.04 | 18.57 | 45.60 |
| | 100th best score | **52.24** | 16.52 | 52.09 | - | 38.11 | 13.58 | 28.43 |
| | 1000th best score | 44.10 | 14.33 | **50.28** | - | - | 6.83 | 17.27 |
| | Unique mutations | 1418 | 1018 | 115 | 18 | 25 | 1743 | **1752** |
| | Unique sites | 330 | 244 | 73 | 13 | 18 | **346** | 345 |
| **Seq2Fitness** | % better than WT | 100 | 100 | 100 | 36.32 | 42.60 | 7.33 | 3.52 |
| | Best score | 5.64 | 5.22 | **5.97** | 4.84 | 5.31 | 4.84 | 3.18 |
| | 100th best score | 5.19 | 4.72 | **5.51** | 0.91 | 1.31 | 0.96 | 0.90 |
| | 1000th best score | 4.88 | 4.03 | **5.20** | - | - | 0.57 | 0.34 |
| | Unique mutations | 655 | 3608 | 393 | 509 | 713 | 3167 | **3629** |
| | Unique sites | 228 | 410 | 181 | 209 | 238 | 423 | **424** |
| **Smoothed Seq2Fitness (w/GGS)** | % better than WT | 49.00 | **72.97** | 33.12 | 41.49 | 44.40 | 8.32 | 3.68 |
| | Best score | 4.70 | **5.07** | 4.39 | 4.84 | 4.84 | 4.84 | 3.18 |
| | 100th best score | 2.43 | **4.15** | 2.95 | 0.92 | 1.23 | 0.97 | 0.86 |
| | 1000th best score | 1.35 | **1.52** | 1.43 | - | - | 0.52 | 0.31 |
| | Unique mutations | 3402 | 2735 | **3955** | 330 | 421 | 2744 | 3480 |
| | Unique sites | 398 | 373 | 414 | 138 | 187 | 423 | **424** |

for ESM2. Importantly, when using a temperature of 0.1 EvoProtGrad does not even find 10,000 unique sequences for this task or NucB. Moreover, applying GGS-smoothing with the Seq2Fitness model, as proposed by [17], did not clearly improve the results of either BADASS or EvoProtGrad sampling. For example, GGS led to a higher proportion of sequences found being better than the wildtype sequence with EvoProtGrad, but a reduction in the same metric with BADASS, and the scores of the best, best 100th and best 1000th sequences relative to the corresponding Seq2Fitness EvoProtGrad or BADASS runs improved for some mutation numbers, remained unchanged in others, and decreased in others. Still, with GGS, BADASS outperformed EvoProtGrad with up to 73% of designed sequences having higher scores than the wildtype for BADASS, but only up to 44% for EvoProtGrad. Additionally, the best scoring sequences found by BADASS consistently achieved higher scores than the best sequences found by EvoProtGrad both with and without the GGS smoothing: BADASS found a sequence with an ESM2 score of 55.91 versus 54.04 for EvoProtGrad, and a sequence with a Seq2Fitness score of 5.97 versus 5.31 for EvoProtGrad.

On the NucB optimization tasks (Tables 3), BADASS similarly showed superior performance, with 100% of the top 10,000 sequences generated achieving higher scores than the reference sequence using both ESM2 and Seq2Fitness models. However, with EvoProtGrad, as few as 12.1% of top 10,000 sequences had higher fitness scores compared with the reference sequence. However, in contrast to alpha-amylase tasks, smoothing with GGS did not negatively affect the fraction of sequences that outperformed the reference sequence for NucB, and GGS led to higher fitness scores achieved by of the best scoring sequence. GGS also improved the performance of EvoProtGrad on both the fraction of high-scoring sequences and the fitness scores of the best sequences. Hence, we conclude that the advantage of GGS-smoothing

**Table 3. NucB sampling:** Performance comparison between BADASS, EvoProtGrad and GGS using ESM2 and Seq2Fitness models. As benchmark, the reference NucB sequence has an ESM2 fitness of 0.0, and a Seq2Fitness score of -0.677. BADASS was run for 200 iterations with a batch size of 520 sequences.

| | | BADASS | | | EvoProtGrad | | | |
| --- | --- | --- | --- | --- | --- | --- | --- | --- |
| | | Gamma = 1 | | | T = 0.1 | | T = 1.0 | |
| Fitness | Metric | k = 2-6 | 2 | 6 | 1-2 | 1-6 | 1-2 | 1-6 |
| **ESM2** | % better than WT | **100** | **100** | **100** | 92.86 | 99.08 | 91.25 | 97.04 |
| | Best score | 45.00 | 16.05 | **45.10** | 11.53 | 40.46 | 14.46 | 32.08 |
| | 100$^{th}$ best score | 41.32 | 13.23 | **41.99** | - | 10.14 | 10.66 | 24.22 |
| | 1000$^{th}$ best score | 33.60 | 11.38 | **39.96** | - | - | 6.38 | 15.90 |
| | Unique mutations | 1200 | 736 | 208 | 11 | 22 | 1256 | **1282** |
| | Unique sites | 128 | 98 | 67 | 5 | 9 | 135 | **136** |
| **Seq2Fitness** | % better than WT | **100** | **100** | **100** | 78.74 | 91.87 | 18.24 | 12.13 |
| | Best score | **6.12** | 3.25 | 5.81 | 2.69 | 4.04 | 2.11 | 3.01 |
| | 100$^{th}$ best score | **5.49** | 2.42 | 5.44 | 0.85 | 2.36 | 0.29 | 0.52 |
| | 1000$^{th}$ best score | 4.82 | 1.87 | **5.12** | - | - | -1.17 | -1.47 |
| | Unique mutations | 380 | 686 | 135 | 404 | 466 | 1620 | **1770** |
| | Unique sites | 77 | 99 | 48 | 83 | 90 | **141** | **141** |
| **Smoothed Seq2Fitness (w/GGS)** | % better than WT | **100** | **100** | **100** | 85.93 | 94.18 | 59.47 | 72.05 |
| | Best score | 5.88 | 3.25 | **6.24** | 2.76 | 4.62 | 2.54 | 3.97 |
| | 100$^{th}$ best score | 5.45 | 2.42 | **5.62** | 0.65 | 2.73 | 0.63 | 1.52 |
| | 1000$^{th}$ best score | 4.81 | 1.86 | **5.26** | - | - | - | -0.68 |
| | Unique mutations | 349 | 642 | 197 | 166 | 231 | 539 | **650** |
| | Unique sites | 75 | **96** | 55 | 56 | 68 | 88 | 94 |

may be dependent on the task and the nature of the fitness landscape of the specific protein under study. On this task, the best ESM2 score BADASS found was 45.10 versus 40.46 with EvoProtGrad, and the best Seq2Fitness score found was 6.24 with BADASS versus 4.62 with EvoProtGrad.

In addition to generating high-scoring sequences, BADASS simultaneously maintained a high level of diversity in the generated sequences despite achieving high fitness values. Paired with the Seq2Fitness model, BADASS identified sequences with a substantial number of unique mutations and mutated sites, representing up to 45% and 25% of possible mutations and 96% and 70% of possible sites in alpha-amylase and NucB respectively. In contrast, EvoProtGrad achieved similar levels of diversity with a high sampling temperature but at the expense of substantially lower fitness scores. EvoProtGrad, as with other MCMC approaches, tends to find sequences with high fitness scores only with a sufficiently low sampling temperature such that the diversity of generated sequences is low, as is evidenced by the number of unique mutations and unique sites. BADASS, however, is able to explore a broad range of sampling temperatures in a single run due to the dynamic temperature control and identify high scoring sequences without loss of diversity. Tables I–L in S1 Text show detailed results comparing BADASS and EvoProtGrad for alpha-amylase and NucB.

Furthermore, we evaluated the contribution of the key features of BADASS, showing in Fig A and Table M in S1 Text that replacing the temperature control based on the average score with a simple cooling schedule, or not updating the mutation energies based on the sequences sampled throughout the optimization, significantly hurts performance, designing only sequences with lower fitness scores. The superior performance of BADASS over state-of-the-art methods is a direct result of its theoretical framework and computational efficiency. Unlike gradient-based methods such as EvoProtGrad and GGS, which rely on computationally expensive gradient evaluations and can converge prematurely on local optima, BADASS operates solely on forward model evaluations. This allows it to explore a significantly larger

sequence space while maintaining diversity. Critically, as shown in Fig 2C, BADASS's biphasic temperature control enables concentrated sampling in regions where the upper envelope of possible scores peaks at intermediate temperatures, precisely where the probability of finding high-fitness sequences is maximized. These results align with the theoretical predictions of BADASS's dynamic temperature control and adaptive mutation energy updates, demonstrating the effectiveness and robustness of the algorithm for protein design.

## Algorithm and theoretical foundation for BADASS optimization

We define the shell of all variants with exactly $k$ mutations away from the reference sequence as $\mathcal{S}_k$. For a protein with $L$ amino acids, this shell contains $\binom{L}{k}19^k$ sequences, a number that grows so quickly with $k$ that enumerating and scoring all sequences in the shell is only

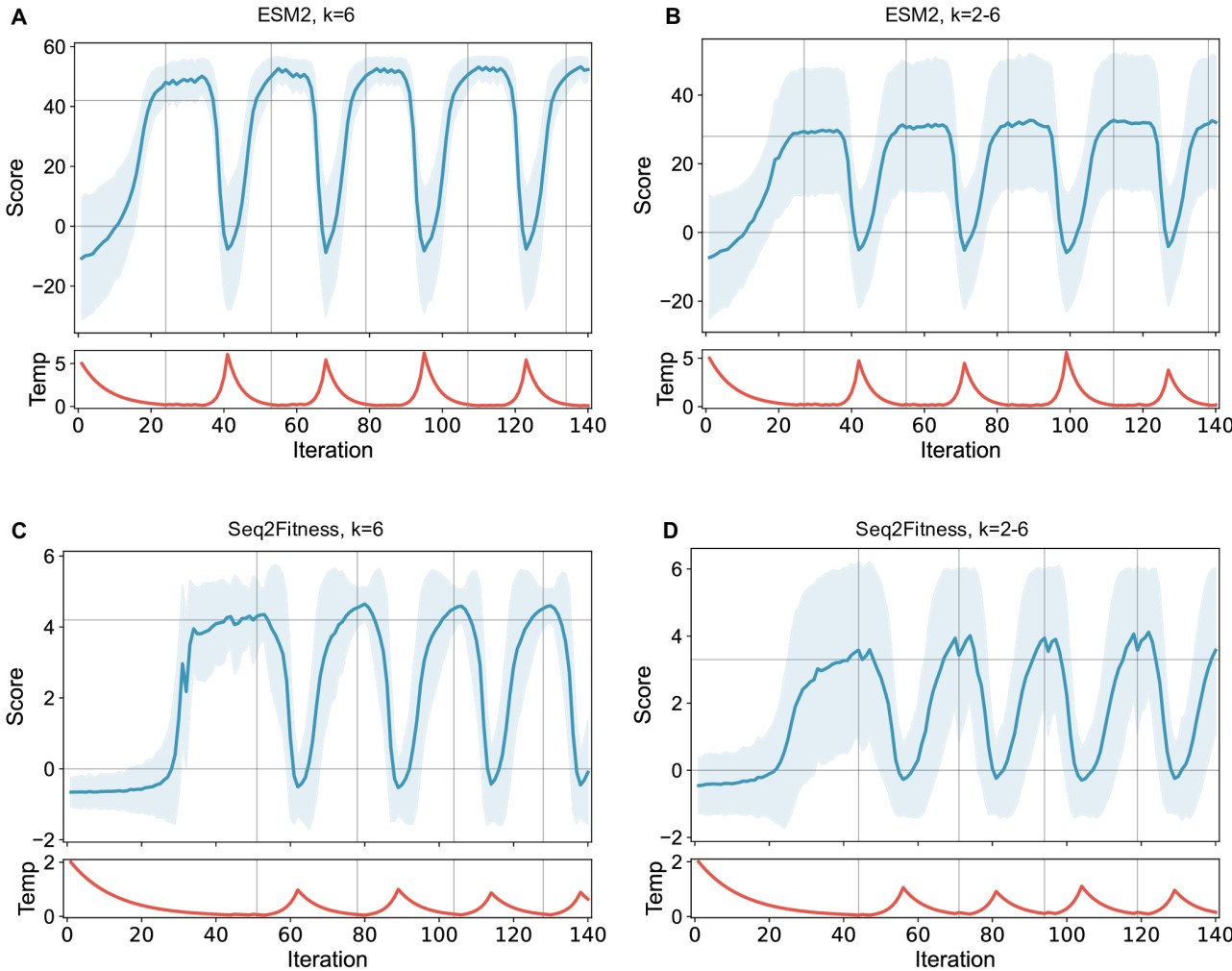

**Fig 2. Fitness score statistics for BADASS optimization of alpha-amylase (AMY_BACSU).** BADASS was run for 140 iterations with a batch size of 1,000. The plot shows fitness score averaged over sampled sequences per iteration, with the shaded area representing scores within $1.96\sigma$ of the average, with $\sigma$ the standard deviation of batch scores. Horizontal lines denote the set points $\mu_{\text{low}}$ and $\mu_{\text{high}}$ that govern the transitions between cooling and heating phases (Fig 1C). Vertical lines mark iterations where phase transitions occur as a moving average of fitness scores crosses the thresholds. The optimization was performed using the unsupervised ESM2 model and the semi-supervised Seq2Fitness model. Fitness scores were standardized as described in the methods. Runs included either exactly 6 mutations per variant (A, C) or an even mix of 2 to 6 mutations (B, D).

practical for $k = 1$ for typical protein lengths of several hundred amino acids. The BADASS algorithm can explore $\mathcal{S}_k$, or several shells at once, searching through the larger set $\mathcal{M}_k$ of sequences with $k$ or fewer mutations. We let $1 \leq m \leq M$ be an index corresponding to the possible single mutations, where $M = L \times 19$. We specify an amino acid sequence $x$ relative to the reference sequence as the set of mutations in $x$. Let $\mathcal{S}_{m,t}$ be the set of sequences sampled and scored up to iteration $t$ that contain mutation $m$, and $\mathcal{X}_t$ be the set of sequences sampled at iteration $t$. We describe BADASS next, and discuss how to set its parameters in the Results section.

1. **Initialization**: Score all $M$ single mutant sequences with the fitness model. A reference fitness score $f_o$ and scale $\tau$ are specified by the user to normalize the scores for numerical stability. We define the normalized energy of mutation $m$ as

$$q_m = \frac{1}{Q} e^{-e_m/T_0}, \text{ where } e_m = \big(f_o - f(x_m)\big)/(1 + \tau) \tag{1}$$

   $x_m$ is the sequence with (single) mutation $m$, $T_0$ is the initial temperature, and $Q = \sum_{m=1}^{M} e^{-e_m/T_{t+1}}$ is the partition function. The sequence set is initialized with the single mutants, $\mathcal{S}_{m,0} = \{f(x_m)\}$. Then, the high and low average score set point values $\mu_{\text{high}}$ and $\mu_{\text{low}}$ are specified by the user; these values guide the temperature updates. Additionally, the base cooling rate $\alpha$, the heating rate $\alpha_{\text{heat}}$, and the accelerated cooling rate $\alpha_{\text{cool}} < \alpha$ are defined. The optimizer state is set to *initial transient*.

2. **Iteration**: Sequences are iteratively sampled until a defined budget is exhausted.

   i. **Sample sequences**: In each iteration, $N$ multi-mutant sequences are sampled from the distribution

$$q_{t+1}(x) \quad = \quad \prod_{m \in x} q_m, \tag{2}$$

   and stored in the set $\mathcal{X}_t$. Each sequence $x$ has $k$ mutations when exploring $\mathcal{S}_k$, or $k$ or fewer mutations according to user-specified proportions when exploring $\mathcal{M}_k$.

   ii. **Score sequences**: Scores for previously sampled sequences in $\mathcal{X}_t$ are retrieved from a cache, while newly sampled sequences are evaluated using the fitness model. These newly scored sequences are then added to the sets $\mathcal{S}_{m,t}$, with each new sequence with $k$ mutations being added to the $k$ sets corresponding to the mutations it contains.

   iii. **Update optimizer state**: The mean and variance of the fitness scores are computed for sequences sampled during the current iteration, as follows:

$$\mu_t = \frac{1}{N} \sum_{x \in \mathcal{X}_t} f(x), \text{ and } \sigma_t^2 = \frac{1}{N} \left( \sum_{x \in \mathcal{X}_t} f^2(x) \right) - \mu_t^2. \tag{3}$$

   The optimizer state is updated based on the mean score. If the simple moving average of $\mu_t$ is greater than $\mu_{\text{high}}$, the optimizer state is set to *active phase transition*. The optimizer remains in this state for a predefined number of iterations (*patience*), or until the score rapidly declines, after which it is switched to *phase transition reversal*. Conversely, if $\mu_t < \mu_{\text{low}}$, the state is changed to *cooling phase*.

   iv. **Update temperature**: The temperature is adjusted based on the current state of the optimizer. If in *initial transient*, the system cools at the base rate, updating the temperature as $T_{t+1} = \alpha T_t$. Cooling is continued during an active phase transition. However, in *phase transition reversal*, the temperature is increased rapidly according to $T_{t+1} =$

$\alpha_{\text{heat}} T_t$. During the *cooling phase*, the temperature decreases quickly, following $T_{t+1} = \alpha_{\text{cool}} T_t$.

 v. **Update sampler** through the following calculations:

 (i). Mean of mutation scores: $\tilde{f}_m = \frac{1}{|\mathcal{S}_{m,t}|} \sum_{x \in \mathcal{S}_{m,t}} f(x)$

 (ii). Variance of mutation scores: $\sigma_{\tilde{f},m}^2 = \frac{1}{|\mathcal{S}_{m,t}|} \sum_{x \in \mathcal{S}_{m,t}} f(x)^2 - \tilde{f}_m^2$

 (iii). Raw mutation energies: $\tilde{e}_m = -\tilde{f}_m - \gamma \sigma_{\tilde{f},m}$. We typically use $\gamma = 1.0$.

 (iv). Normalized mutation energies: $e_m = (\tilde{e}_m + f_o)/(1 + \tau)$

 (v). New mutation probabilities: $q_m = \frac{1}{Q} e^{-e_m/T_{t+1}}$.

3. **Terminate**: Once the budget is met, the resulting sequences are analyzed to select the desired ones. The simplest method ranks sequences by their scores and retains the top-ranking sequences. However, more complex selection criteria can be applied, such as limiting the number of sequences with specific mutations or targeting specific mutation sites. In this work, we simplify by retaining only the highest-scoring sequences.

Algorithms A-C in Appendix C of S1 Text summarize BADASS in pseudo-code for easy reference. After an initial transient of tens of iterations, BADASS settles in fairly regular oscillations (e.g., see Fig 2) that trace out clear patterns in the mean and variance of the model score versus temperature. Figs 3 and 4 show these traces for the four tasks we consider, along with fits to equations we developed to explain the behavior. For the ESM2 tasks and when sampling sequences with a fixed number of mutations, the change in $\mu_t$ and $\sigma_t^2$ with temperature is monotonic. But for the ML model tasks or the ESM task when we sample a blend of sequences with different numbers of mutations, the variance $\sigma_t^2$ shows an interesting peak at intermediate temperatures. In Appendix D in S1 Text, we develop a simple model to understand these behaviors, and state the equations we use to fit the data in Figs 3 and 4.

Unlike algorithms that iteratively update the reference by accumulating mutations over generations, BADASS maintains a fixed reference sequence (typically the wildtype) throughout the optimization process. This approach enables efficient exploration of the k-mutation neighborhood around a single reference point, rather than stepwise mutation accumulation. The optimization process expands the number of sampled sequences with each iteration while maintaining the same reference point. The biphasic structure and dynamic mutation energies in BADASS are motivated by theoretical considerations and empirical observations. The dynamic mutation energies are continuously refined using sampled sequences, with high-temperature phases providing less biased samples that improve these estimates despite their lower scores. Low-temperature phases then exploit these improved estimates to find high-scoring sequences. This alternation is crucial - neither pure cooling (which can prematurely converge) nor static mutation energies (which fail to incorporate new information) achieve comparable performance. Importantly, our empirical results show that intermediate temperatures maximize the upper envelope of possible scores (Fig 2C), suggesting an optimal trade-off between exploration and exploitation. This observation aligns with our theoretical analysis showing that the variance of scores peaks at intermediate temperatures, making the biphasic approach particularly effective at maintaining diversity while finding high-fitness sequences.

Next, we motivate BADASS from theory, and study aspects of its convergence to an ideal but impractical sampler. The success of BADASS is grounded in its theoretical design, which leverages principles from statistical mechanics to balance exploration and exploitation in

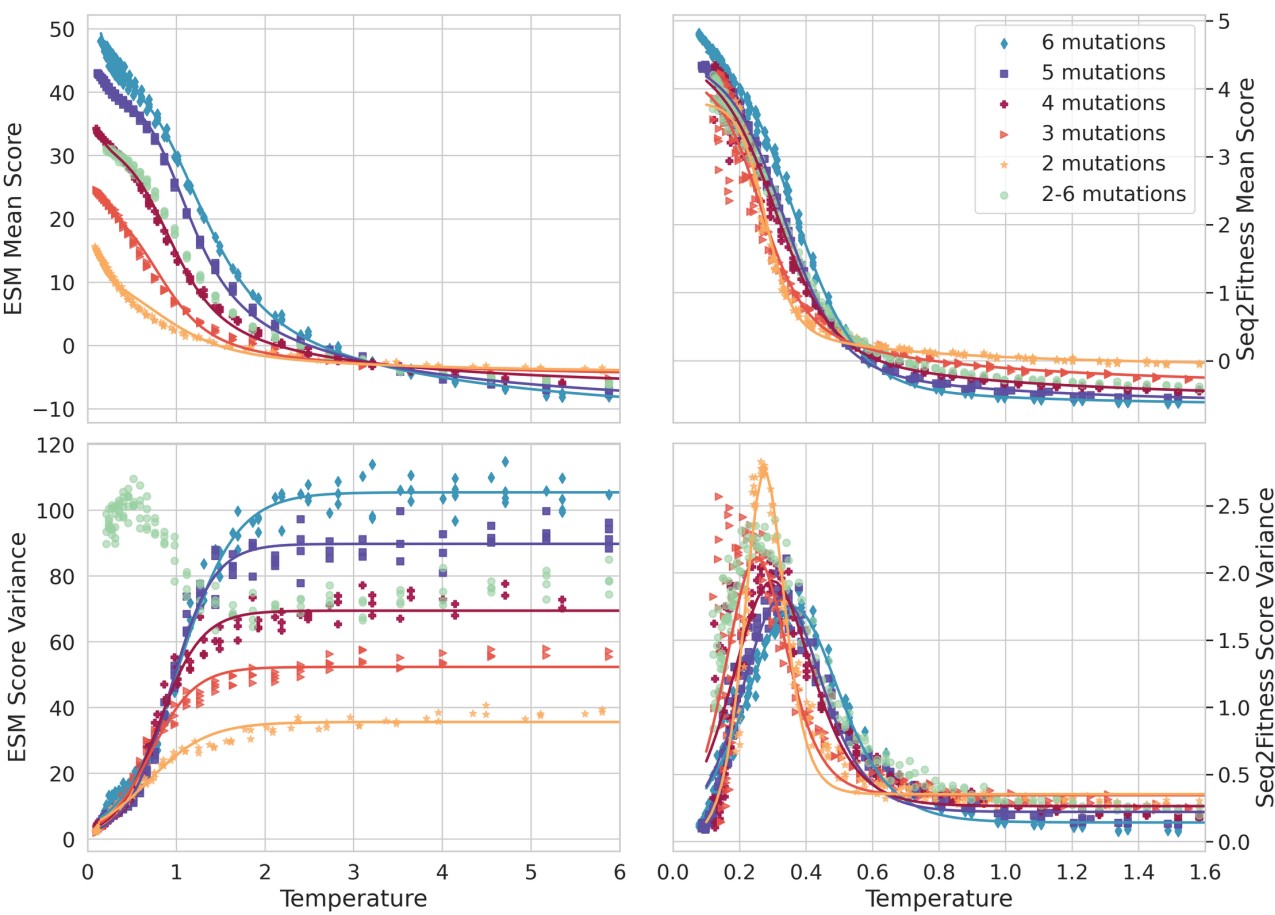

**Fig 3. Order parameters versus temperature for amylase tasks:** mean score and variance of scores versus temperature after initial transient, i.e., at steady-state oscillatory BADASS behavior. Markers come from BADASS runs, and lines are fits using Eqs 5–7 in S1 Text. These were obtained from cooling then heating runs of our algorithm for the amylase task: on the left using the ESM2 mutant marginal score, and on the right using the machine learning model that predicts fitness for stain removal and dp3 function. We ran the algorithm for 250 iterations, scoring 500 sequences in each iteration, and show all data for iterations larger than 100 to avoid the initial transient. The peak of the variance at intermediate temperatures is striking. Running the algorithm with an even blend of numbers of mutations changes the variance behavior, and was not fit to our equations. The mean and variance traces here are reminiscent of the magnetization and susceptibility in Ising models.

sequence space. Here, we provide a mathematical framework that explains how BADASS sustains diversity while optimizing for high-fitness sequences. By understanding this foundation, we contextualize the algorithm's experimental performance where its ability to avoid premature convergence and find high fitness proteins as the temperature changes becomes evident. To effectively sample diverse, high-scoring sequences, we seek a probability mass function (simply referred to as a distribution going forward) over the sequence space $\mathcal{S}_k$. Defining $\mathcal{P}_k$ as the space of such distributions, a natural problem to solve is:

$$p(x) = \arg\max_{p' \in \mathcal{P}_k} \mathcal{L}(p'), \text{ where } \mathcal{L}(p') = \langle f \rangle_{p'} + TH(p').  \tag{4}$$

Throughout this work, angled brackets denote averages, with the probability distribution used for the average as a subscript, though omitted when clear from context. So, e.g., $\langle f \rangle_{p'} = \sum_x p'(x)f(x)$. In Eq 4, $H(p') = -\langle \log p'(x) \rangle_{p'}$ is the Shannon entropy of $p'(x)$, and the

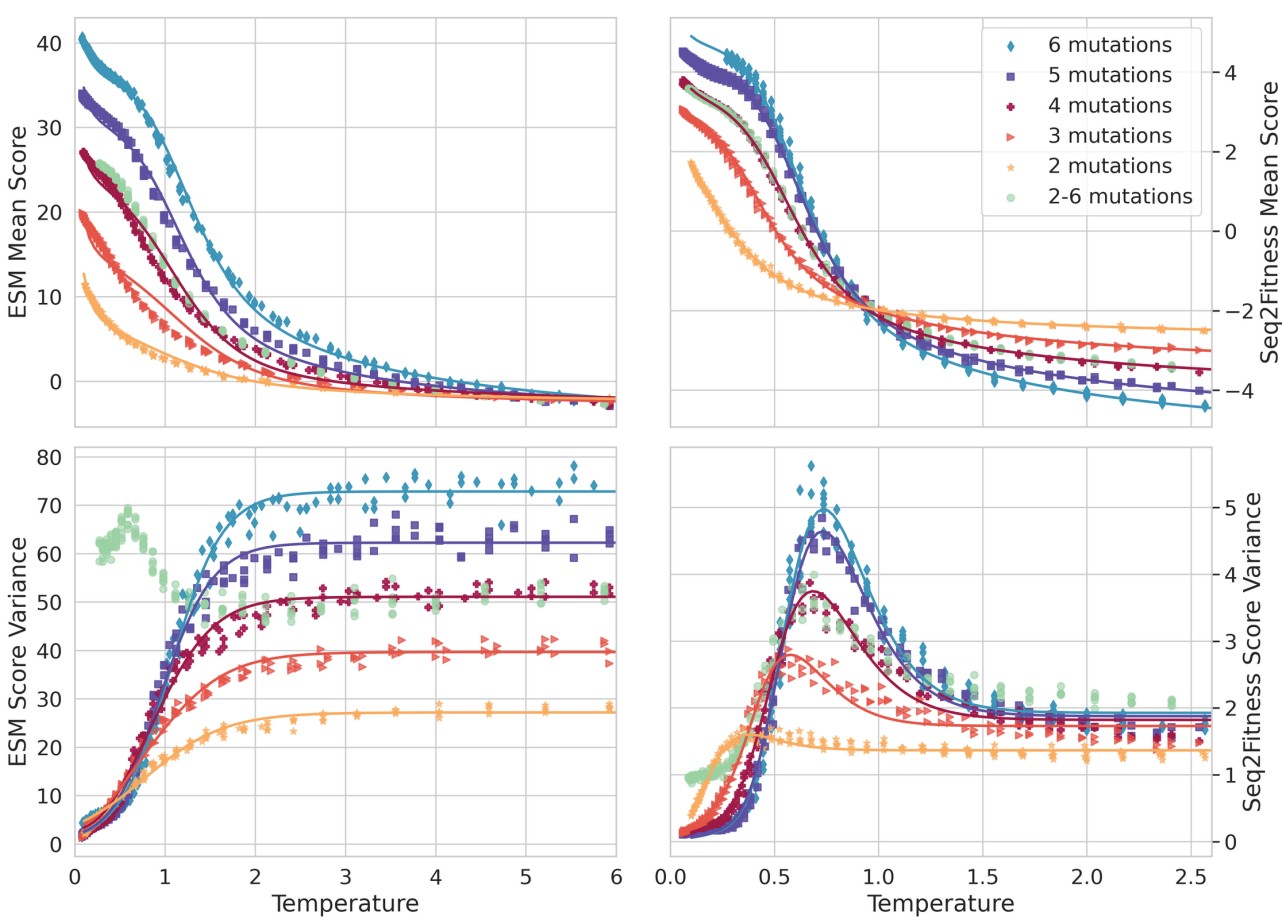

**Fig 4. Order parameters versus temperature for the NucB tasks:** analogous to Fig 3. Left corresponds to ESM2 mutant marginal scores, and right the Seq2Fitness model score: the logit for the probability that the nuclease activity is higher than that of the reference sequence.

temperature parameter $T \geq 0$ controls the trade-off between maximizing the average fitness score and the diversity of $p'(x)$. The well-known solution to Eq 4 is the Boltzmann distribution with energy $E(x) = -f(x)$:

$$p(x) = \frac{1}{Z}e^{f(x)/T}, \text{ where } Z = \sum_{x \in \mathcal{S}_k} e^{f(x)/T} \tag{5}$$

is the partition function. However, because the partition function is the sum of a large number of terms, each requiring evaluation of $f(x)$, direct computation of $Z$ and $p(x)$ is intractable. Recent approaches enable reasonably effective MCMC sampling from Eq 5 by defining a tractable proposal distribution with high acceptance probabilities. Specifically, Evo-ProtGrad [16] introduced a proposal distribution based on the gradients of $f(x)$ with respect to the amino acids as one-hot encodings, and a subsequent work called GGS [17] used the same proposal distribution, but applied to a modified model trained on an augmented and smoothed dataset to avoid local optima. We compared these approaches to BADASS in our results.

Our approach is different from MCMC. We restrict the space of probability distributions to a smaller subspace $\widetilde{\mathcal{P}}_k \subset \mathcal{P}_k$ of distributions of the form $q(x) = \prod_{m \in x} q_m$, where $k$ mutations are sampled independently from each other from a distribution over mutations with entries $q_m \geq 0$, where $1 \geq m \geq M$ indexes the possible substitutions and $\sum_{m=1}^{M} q_m = 1$. Sampling from any distribution in $\widetilde{\mathcal{P}}_k$ is straightforward, and we can find one that is close enough to Eq 5 to yield the diverse and high-scoring sequences we seek. To choose the distribution in $\widetilde{\mathcal{P}}_k$, a logical direction is maximizing the same objective as before over the more restricted space, i.e., $\mathcal{L}(q)$. Rewriting this objective in terms of $q_m$ and calculating its gradient with respect to $q_m$ results in non-linear equations that do not appear to have a closed form solution. But these gradients can be used to optimize $\mathcal{L}(q)$ numerically with respect to $q_m$, e.g., via Stochastic Gradient Descent as shown recently by [20]. Instead of minimizing $\mathcal{L}(p)$ directly over $\widehat{\mathcal{P}}_k$, we find the distribution in $\widehat{\mathcal{P}}_k$ that best approximates the Boltzmann distribution of Eq 5 in terms of the Kullback-Leibler divergence:

$$q^*(x) = \arg\min_{q' \in \widehat{\mathcal{P}}_k} D_{\text{KL}}(p \| q') = \prod_{m \in x} q_m^*, \text{ where } q_m^* = \frac{1}{kZ} \sum_{x \in \mathcal{N}_m} e^{f(x)/T}. \tag{6}$$

Here we let $\mathcal{N}_m$ be the set of all sequences with $k$ mutations that contain mutation $m$. Substituting $q^*(x)$ into $D_{\text{KL}}(p \| q)$ yields the non-trivial inequality $kH(q_m^*) \geq H(p)$ that provides an upper bound of the entropy of the Boltzmann distribution in Eq 5 in terms of the the entropy of the optimal distribution over mutations $q_m^*$. It is valid for any positive integer $k$, although $p(x)$ and $q_m^*$ also depend on $k$; the former through its support, and the latter through the sets $\mathcal{N}_m$.

We solve the optimization problem in Eq 6 to obtain $q_m^*$ in Appendix E in S1 Text. Because $\mathcal{N}_m$ contains $n_k = \binom{M-1}{k-1}$ sequences, the entries $q_m^*$ in Eq 6 cannot be computed in practice, so we make two final approximations and one modification. First, we interpret the summation as proportional to the average of $e^{f(x)/T}$ over a uniform distribution over the sequences in $\mathcal{N}_m$. We could approximate this average with samples of $x \in \mathcal{N}_m$ but the result is statistically inefficient with high variance because of the exponential function, and it leads in practice to poor samplers. So, instead we pursue the mean field approximation

$$\tilde{q}(x) = \prod_{m \in x} \tilde{q}_m = \frac{1}{\tilde{Q}} e^{f_m/T}, \text{ where } f_m = \frac{1}{n_k} \sum_{x \in \mathcal{N}_m} f(x) \tag{7}$$

is the average score of all sequences with mutation $m$, and $\tilde{Q} = \sum_m e^{f_m/T}$ is the partition function. Next, we study how $D_{KL}(p \| q^*)$ and $D_{KL}(p \| \tilde{q})$ change as the system is cooled by decreasing the temperature. We find that

$$\partial_\beta D_{KL}(p \| q^*) = \beta \sigma_f^2 - k\text{Cov}_{q_m^*}\left(\langle f | m \rangle_p, \log q_m^*\right), \text{ and} \tag{8}$$

$$\partial_\beta D_{KL}(p \| \tilde{q}) = \beta\left(\sigma_f^2 - k\text{Cov}_{q_m^*}\left(\langle f | m \rangle_p, f_m\right)\right) - k\left(\langle f_m \rangle_{q_m^*} - \langle f_m \rangle_{\tilde{q}_m}\right), \tag{9}$$

where $\beta = 1/T$ is the inverse temperature, $\langle f | m \rangle_p$ is the average sequence score when sampling sequences from $p(x)$ conditioned on the set containing mutation $m$, and $\sigma_f^2$ the variance of sequence scores under $p(x)$. These equations are some of our main results, and are derived in Appendix F in S1 Text. For the KL divergences to decrease as the system cools, the covariance terms have to be positive enough for the entire expression to be negative. This can be checked approximately for low and for high temperatures.

We find that at high temperatures, these two KL divergences change approximately at the same small rate proportional to $1/T$, and have an ambiguous sign. At low-enough temperatures, however, Eq 8 approximately becomes

$$\partial_\beta D_{KL}(p\|q^*) = \beta(1-k)\sigma_f^2,  \tag{10}$$

which is negative (assuming a non-zero sequence variance) for $k>1$, e.g., for sequences with 2 or more mutations, and proportional to $1/T$. Furthermore, the magnitude increases with $k$ as long as $\sigma_f^2$ does not decrease faster than linearly with $k$ (intuitively, we expect this variance to in fact to grow as $k$ increases), predicting a sharper improvement in the closeness between $q^*(x)$ and $p(x)$ as the system is cooled for sequences with an increasing number of mutations, consistent with our empirical results in Figs 3 and 4, and in the amylase and NucB tables of BADASS results in Appendix B in S1 Text. In the case of the mean field approximation $\tilde{q}$, at low enough temperatures,

$$\partial_\beta D_{KL}(p\|\tilde{q}) \approx \beta\left(\sigma_f^2 - k\mathrm{Cov}_{q_m^*}\left(f_m^*, f_m\right)\right) - k\left(f^* - \max_m f_m,\right)  \tag{11}$$

where $f^*$ is the maximum score over all sequences, and $f_m^* = \max_{x\in\mathcal{N}_m} f(x)$ is the largest score of all sequences with mutation $m$. When $\mathrm{Cov}_{q_m^*}(f_m^*, f_m)$ is positive, the KL distance will become smaller as the system cools if the magnitude of the covariance times $k$ exceeds $\sigma_f^2$, therefore with improved convergence as $k$ increases. The low and high temperature approximations of Eqs 8 and 9 are developed in Appendices G and H in S1 Text, respectively. To get to our sampling distribution for BADASS, our second approximation is to estimate $f_m$ using only the sequences we have scored so far that have mutation $m$. This induces a complex bias in our estimate that is optimization-path dependent, but we find that the resulting sampler works well in practice. But the empirical results are often stronger when we encourage the sampling of mutations that have a high standard deviation of scores, in addition to having a large average score, resulting in the modification that yields our final distribution for our sequence sampler defined in step 2v of our algorithm; e.g., compare results in Tables I and K in S1 Text for $\gamma = 0$ and $\gamma = 1$.

## Discussion

We introduced Seq2Fitness, a semi-supervised method for predicting protein fitness from experiment and evolutionary data, and BADASS, a biphasic annealing algorithm that updates mutation energies as it processes sequences. In comparison with selected state-of-the-art fitness prediction methods, Seq2Fitness showed superior performance, particularly demonstrating improved ability to extrapolate to new mutations and positions. From ablations, we showed that the unique combination of relative embeddings, logits, normalized zero-shot scores with parallel path convolutions enabled Seq2Fitness to better extrapolate in the fitness landscape compared to other models. Our application of Seq2Fitness aimed to maximize predictive performance using semi-supervised regression with frozen language model representations, without backpropagating gradients to update language model parameters. Instead, we relied on the language model to generate per-residue embeddings, logits, and scores and fed these to a top model for further training. However, recent advances in parameter-efficient fine-tuning, such as LoRA [33], have shown improved performance compared to training protein regression models on frozen representations [34,35] For future research, we highlight

that these fine-tuning approaches can be easily integrated into our semi-supervised framework and combined with our biphasic optimizer for protein design. Furthermore, as an additional layer of screening for improved designs, structural evaluations such as pLDDT scores or force-field-based stability predictions like FoldX or Rosetta could be applied to evaluate proteins designed by BADASS [36–38] However, these methods may incur substantial computational costs.

We evaluated BADASS and demonstrated that it consistently outperforms alternative methods for sampling high-scoring protein sequences. While the focus of this work has been on optimizing amino acid sequences, BADASS could be applied to explore other biological sequence spaces, such as DNA, RNA, and even synthetic molecules like lipids and polymers. Additionally, BADASS could be extended to handle more complex mutation types, including insertions and deletions, or multi-output models where multiple properties (e.g., reaction conversion and specificity) must be optimized simultaneously.

While the recently published optimization method, MODIFY [20], shares some theoretical similarities with BADASS, MODIFY samples mutations from a small subset of carefully selected sites. For design of protein variants where mutations at every site are considered, we observed that MODIFY collapsed and resulted in sequences with substantially lower fitness scores, often failing to identify any sequence with a higher score than the reference. Hence, we focused our comparisons on optimization methods that are designed to handle mutations at all sites.

It is important to note that the current implementation of Seq2Fitness requires protein-specific training for each reference sequence of interest. Unlike zero-shot models such as ESM2, which can be applied to any protein sequence without additional training, Seq2Fitness models are trained specifically for each protein family being studied, which enables higher prediction accuracy at the cost of reduced generalizability. A major limitation affecting our work, and the broader protein engineering field, is the scarcity of high-quality, large-scale datasets involving multiple mutations, particularly those with detailed functional annotations. Most available datasets focus on single mutations, which limits our ability to train models that generalize to sequences with multiple mutations. Additionally, existing datasets often lack essential metadata, such as experimental conditions like temperature and pH, which is critical for training models that perform consistently across different contexts. To enable further advances in protein design, the field should invest in generating and sharing annotated, multi-mutant datasets across diverse protein families and functions [39,40]. Collaborative efforts among academic institutions, biotech companies, and public agencies will be crucial in creating these resources. As more standardized multi-mutant datasets become available, they will enable robust benchmarking and the continued development of optimization algorithms like BADASS. To facilitate reproducibility, we have made our code available, encouraging the community to adapt it for their own research.

## Materials and methods

### Seq2Fitness architecture and training

The Seq2Fitness model uses per-residue embeddings from the final transformer layer of the ESM2-650M protein language model to provide rich representations of protein sequences [22]. For each variant, we computed relative embeddings by subtracting the wildtype embedding. We also retrieved logits from the ESM2 output layer and transformed them into log probabilities, resulting in two matrices per sequence: an embedding matrix of size, $L \times 1280$, and a log probability matrix of size, $L \times 20$, where L is the sequence length. Each of these

matrices were fed to two parallel convolutional paths (Dual-CNN). One path applies a convolution layer followed by a percentile-based statistical summary across the sequence, while the other computes statistical summaries before convolution. Zero-shot fitness scores, including wildtype and mutant marginal scores from ESM2-650M, and masked marginal scores from ESM2-3B [22], as defined by Meier et al [7], were computed as unsupervised fitness predictions. To correct bias in unsupervised scores across variants with varying number of mutations relative to the wildtype [11], we computed normalized scores by dividing the scores by the number of mutations, and also passed the number of mutations as a feature. The outputs from the convolutional layers and the fitness scores were concatenated and fed to a multi-layer perceptron with two hidden layers. In cases where fitness labels are not comparable, due to variations in screening conditions or assay types, we used multi-task learning with separate outputs for distinct labels and computed the loss as a weighted sum of per-task losses, with identical weights and losses (e.g., mean squred error or cross entropy, depending on the task nature). We standardized fitness labels in the training set to have zero mean and unit variance for numerical stability. We used hyperparameters that included a filter size of 32 for both convolution paths in each Dual-CNN block (Fig 1B), kernel size of 1, and 11 percentiles for the statistical summaries: 1, 2.5, 12.5, 25, 37.5, 50, 62.5, 75, 87.5, 97.5, and 99. The MLP consisted of two layers with 27 and 15 units, respectively, using the GeLU activation. To mitigate overfitting, we applied a dropout rate of 0.2 and weight decay of 2e-3. The model was trained with an initial learning rate of 1e-2 and a cosine annealing schedule.

## Comparing Seq2Fitness with alternative methods

We compared Seq2Fitness with selected state-of-the-art fitness prediction methods from the literature. First, we evaluated zero-shot predictions from ESM2, computed by averaging wildtype and mutant marginal scores as defined in [7]. Additionally, we trained an L2-regularized linear model (ridge) with one-hot representations of the proteins (Linear one-hot) and with mean-pooled embeddings from the ESM2 model (Linear ESM) [23]. Following Hsu et al, we also trained augmented ESM models by concatenating one-hot encodings with mutant and wildtype marginal zero-shot scores (Aug. ESM) [10]. For supervised CNN models, we used the architecture proposed by Gelman et al that is most similar to ours (cnn-1xk3f32), which has a kernel size of 3 with 32 filters, and a single-hidden layer MLP with 100 units [9,18]. While their model featurized proteins using one-hot encodings and the top 20 principal components of amino acid index encodings (AAindex) [41], we trained CNNs using this scheme (CNN AAindex), as well as CNNs using only one-hot encoding (CNN one-hot), and per-residue ESM2 embeddings in place of AAindex (CNN ESM).

The performance of the models was evaluated using the Spearman correlation coefficient, which measures the monotonic relationship between predicted and actual values. This metric is ideal for assessing the model's ability to rank high-fitness variants accurately, focusing on the correct order rather than exact scores. For the NucB dataset with categorical fitness values, we used the Area Under the Receiver Operating Characteristic Curve (AUC), which evaluates the model's ability to distinguish between classes across thresholds. We applied the adjusted AUC, ranging from -1 to 1, to align it with Spearman correlation. For overall comparison across all datasets, the performance values were averaged to derive a single score.

## Choosing parameters for BADASS

Selecting the right parameters for BADASS involves balancing the robustness of fitness score estimates (which improve with a larger batch size), the frequency of sampling distribution updates (once per iteration, so smaller batch sizes directly imply more sampler updates given

a fixed evaluation budget), and GPU utilization when large models like ESM2 or Seq2Fitness are used. Typically, for the protein design tasks discussed here, sampling 500–1000 sequences per iteration strikes a good balance. Our code automatically distributes model inference across available GPUs through PyTorch's DataParallel, enabling these batch sizes and larger ones if desired for ESM2 and Seq2Fitness models on proteins with hundreds of amino acids. A base cooling rate, $\alpha$, in the range of 0.87–0.94 works well, with heating and accelerated cooling rates, $\alpha_{\text{heat}}$ and $\alpha_{\text{cool}}$, set at 1.3–1.8 and $\alpha^3$ or $\alpha^2$, respectively. These values help prevent premature convergence while maintaining diversity in the search process. We generally set the reference score $f_o$ to the 80th percentile and set $\tau$ as the standard deviation of single mutant scores for scaling. Using $\gamma = 1$ leads to robust results; choosing $\gamma = 0$ can sometimes produce better sequences, but it can also sometimes do poorly, getting stuck in a subspace with mediocre sequence scores. BADASS typically stabilizes after 60–100 iterations, showing steady oscillations in mean score and variance after that. Running the algorithm for 200–300 iterations often yields a diverse set of high-scoring sequences. With these choices, it takes 15-40 minutes for a single BADASS run on a machine with two NVIDIA RTX4090 GPUs on the design tasks discussed here for budgets between 100,000 and 300,000 sampled sequences. If needed, simulated annealing or an alternative cooling-heating strategy can be accessed via flags in the code, helping to set the score thresholds $\mu_{\text{high}}$ and $\mu_{\text{low}}$ for cooling and heating phases. For different mutation counts, the same general parameter choices tend to work, simplifying tuning across various tasks. The specific parameter values used in this work are available in the code repository.

## Evaluating BADASS

**Datasets:** We conducted protein optimization tasks on two protein families: the alpha-amylase (AMY_BACSU) dataset and the NucB dataset, where fitness is the measured substrate conversion of a key enzymatic reaction. The alpha-amylase dataset contained 10,722 unique sequences, each 425 amino acids in length, while the NucB dataset included 55,760 sequences, each with 142 amino acids. These datasets contain multi-mutant sequences, and were chosen to reflect a range of protein sizes and sequence complexity. For both protein families, we evaluated designed sequences containing 2–6 mutations relative to a reference sequence.

**Tasks:** Each optimization task involved finding high-scoring sequences based on two fitness models:

1. **ESM2 Model**: a zero-shot model for scoring protein sequences based on their unsupervised representations. We use the ESM2 650M model, and the mutant marginal score as the fitness.
2. **Seq2Fitness Model**: a semi-supervised model trained to predict protein function based on experimental data. This model was trained separately for the amylase and NucB tasks. Since fitness labels for NucB are binary, we used the raw logits before sigmoid activation as the optimization metric (i.e. log-probabilities of activity_greater_than_wt labels).

**Metrics:** We compared BADASS to two other sequence optimization methods: EvoProt-Grad and GGS. The metrics used to evaluate performance include:

1. **Percentage of sequences better than the wild type**: The proportion of the top 10,000 sequences that improved upon the reference sequence's score.

2. **Best, best 100th, and best 1,000th sequence scores**: Fitness scores of the sequences with ranks 1, 100 and 1,000. Sequences with these ranks or higher would be the prime candidates for wet lab validation.

3. **Unique mutations and mutated sites**: The total number of distinct mutations and sites mutated across the top sequences, reflecting each method's ability to maintain diversity in the sequence space.

We compare BADASS against two approaches, described next.

**EvoProtGrad** is the original gradient-based Markov Chain Monte Carlo (MCMC)-based method that generates candidate sequences by iteratively proposing mutations and accepting them based on their impact on sequence fitness [16]. EvoProtGrad did better than other competing approaches, but it found far fewer high-fitness sequences than BADASS, particularly in the NucB tasks where it showed poor coverage of sequence space.

**GGS** is a highly intensive, gradient-based MCMC method designed for protein optimization [17] that aims to avoid locally optimal sequence regions by training a new model on an augmented and smoothed dataset. It explores the sequence space by applying small, iterative mutations, following gradients of the fitness function similarly to EvoProtGrad. But GGS also needs to (i) augment and smooth the dataset, (ii) train a sequence to fitness model on the new larger dataset, (iii) use the resulting model to sample sequences with the gradient-based MCMC that is equivalent to EvoProtGrad, and (iv) re-score the sampled sequences with the original model. Because of code compatibility issues, we used EvoProtGrad code in the sequence sampling step for GGS rather than the GGS code. The result is mathematically equivalent to the GGS procedure, with the exception of an additional clustering-based sequence pruning step that we omit. Despite its intensive nature, the results from GGS were not clearly better than BADASS or EvoProtGrad on the original un-smoothed model.

All optimization methods in our comparison (BADASS, EvoProtGrad, and GGS) were evaluated using the same fitness functions for both optimization and evaluation, which is standard practice in directed evolution optimization. BADASS demonstrates superior performance regardless of whether the fitness function is Seq2Fitness or ESM2, indicating that its effectiveness is not tied to any particular scoring model but rather stems from its efficient exploration of sequence space.

## Supporting information

**S1 Text**. **Supporting Information for this article, including detailed performance results of the Seq2Fitness model across multiple datasets and data splits (Appendix A); extended benchmarking and ablation studies for BADASS and baseline methods (Appendices B–D); theoretical derivations, including BADASS pseudocode, temperature-dependent behavior, convergence analysis, and derivation of key sampling distributions (Appendices C–H); and detailed derivations supporting the theoretical claims in the main text (Appendices E–H).**
(PDF)

## Acknowledgments

We are grateful to Solugen for supporting this work. Yuriy Roman-Leshkov carefully read a draft of this work, and provided valuable suggestions. We thank Patrick Emami for insightful discussions about the EvoProtGrad approach and code.

## Author contributions

**Conceptualization:** Carlos A. Gomez-Uribe, Japheth Gado, Meiirbek Islamov.

**Data curation:** Carlos A. Gomez-Uribe, Japheth Gado, Meiirbek Islamov.

**Formal analysis:** Carlos A. Gomez-Uribe, Japheth Gado, Meiirbek Islamov.

**Investigation:** Carlos A. Gomez-Uribe, Japheth Gado, Meiirbek Islamov.

**Methodology:** Carlos A. Gomez-Uribe, Japheth Gado, Meiirbek Islamov.

**Project administration:** Carlos A. Gomez-Uribe.

**Resources:** Carlos A. Gomez-Uribe.

**Software:** Carlos A. Gomez-Uribe, Japheth Gado, Meiirbek Islamov.

**Supervision:** Carlos A. Gomez-Uribe.

**Validation:** Carlos A. Gomez-Uribe, Japheth Gado, Meiirbek Islamov.

**Visualization:** Carlos A. Gomez-Uribe, Japheth Gado, Meiirbek Islamov.

**Writing – original draft:** Carlos A. Gomez-Uribe, Japheth Gado, Meiirbek Islamov.

**Writing – review & editing:** Carlos A. Gomez-Uribe, Japheth Gado, Meiirbek Islamov.

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
