## [Decision Letter · Decision Letter 0]

PCOMPBIOL-D-24-01706

Designing diverse and high-performance proteins with a large language model in the loop

PLOS Computational Biology

Dear Dr. Gomez-Uribe,

Thank you for submitting your manuscript to PLOS Computational Biology. After careful consideration, we feel that it has merit but does not fully meet PLOS Computational Biology's publication criteria as it currently stands. Therefore, we invite you to submit a revised version of the manuscript that addresses the points raised during the review process.

Please submit your revised manuscript within 60 days (February 09, 2025). If you will need more time than this to complete your revisions, please reply to this message or contact the journal office at ploscompbiol@plos.org. Please include the following items when submitting your revised manuscript:

We look forward to receiving your revised manuscript.

Kind regards,

Mohammad Sadegh Taghizadeh, Ph.D.

Academic Editor

PLOS Computational Biology

Stacey Finley

Section Editor

PLOS Computational Biology

Feilim Mac Gabhann

Editor-in-Chief

PLOS Computational Biology

Jason Papin

Editor-in-Chief

PLOS Computational Biology

**Additional Editor Comments :**

All reviewers raise the points about comparison to other methods / evaluating the performance of this approach. These comments should be addressed.

**Journal Requirements:**

1) We ask that a manuscript source file is provided at Revision. Please upload your manuscript file as a .doc, .docx, .rtf or .tex. If you are providing a .tex file, please upload it under the item type LaTeX Source File and leave your .pdf version as the item type Manuscript.

3) Please upload a copy of Figures 6, 7, and 8 which you refer to in your text on page 10. Or, if the figures are no longer to be included as part of the submission please remove all reference to it within the text.

4) We notice that your supplementary Figures, Tables, and information are included in the manuscript file. Please remove them and upload them with the file type 'Supporting Information'. Please ensure that each Supporting Information file has a legend listed in the manuscript after the references list. Please renumber the figures and tables after removing the supplementary Figures and Tables from the manuscript.

5) Thank you for stating that "Code will be made available at the link provided in the draft." We strongly recommend all authors decide on a data sharing plan before acceptance, as the process can be lengthy and hold up publication timelines. Please note that, though access restrictions are acceptable now, your entire data will need to be made freely accessible if your manuscript is accepted for publication. This policy applies to all data except where public deposition would breach compliance with the protocol approved by your research ethics board. If you are unable to adhere to our open data policy, please kindly revise your statement to explain your reasoning and we will seek the editor's input on an exemption. Please be assured that, once you have provided your new statement, the assessment of your exemption will not hold up the peer review process.

**Reviewers' comments:**

Reviewer's Responses to Questions

Reviewer #1: 1) What does “in the loop” mean in the title of the manuscript?

2) When evaluating the performance of the fitness predictor, the authors compared Seq2Fitness with the very basic methods. Why not use advanced methods as baselines to compare? Such as ECNet, MODIFY and so on.

3) The organization of the Results section seems strange. It is better to remove the sections of “The BADASS algorithm” and “Theoretical motivation for BADASS” from the Results section.

4) The manuscript lacks depth to discuss why the proposed method perform better than the state-of-the-art methods.

Reviewer #2: The article titled “Designing diverse and high-performance proteins with a large language model in the loop” presents and innovative approach to protein engineering. The manuscript introduces BADASS, a novel biphasic annealing algorithm, and Seq2Fitness, a semi-supervised neural network for computational protein engineering. BADASS outperforms optimization methods like MCMC-based EvoProtGrad in discovering high-fitness proteins by adjusting mutation energies and temperatures during optimization. The manuscript provides a novel strategy, and is well-written and organized.

While the approach is innovative, improving the clarity of the methodology could enhance its impact. The biphasic annealing process should be clearer. The authors should consider adding a flow diagram to illustrate the optimization process better. This could also help to differentiate BADASS from MCMC methods for exploring the fitness landscape. Furthermore, the authors could add details to explain why they chose a biphasic structure and dynamic mutation energies. How do these choices balance exploration and convergence of the fitness landscape? Furthermore, the authors could add more information about the evaluation metrics used in the manuscript. The Spearman correlation and adjusted AUC could be defined briefly. The authors could explain why they are well suited to the manuscripts aims. The authors could also add additional information about EvoProtGrad and GGS, and what their strengths and limitations are when applied to the exploration of the fitness landscape. The authors could add additional information about why BADASS achieved better scores than those algorithms, instead of performance statistics. How does the BADASS algorithm improve the “trap-prone” nature of other algorithms? The authors could include some more real-world or experimental validation of the proteins designed with their system in the manuscript to bolster its impact further.

Overall, the manuscript presents a novel approach to computational protein engineering, with advancements over existing methods. Enhancing the clarity of the methodology and providing additional context around the evaluation metrics, algorithmic choices, and performance would improve its impact. Incorporating experimental validation would make this manuscript even more robust. The work has the potential to become a valuable reference.

Reviewer #3: Recap of paper

• The paper a new innovative approach to protein optimization by integrating semi supervised machine learning into a pLM (Seq2Fitness) as well as an annealing strategy for promoting exploration of the fitness landscape (BADASS).

Strengths

• pLM fitness optimization: The Seq2Fitness model uses embeddings from ESM2 to predict fitness landscapes. It combines evolutionary data with experimental labels, and allows extrapolation vs interpolation from training data. It tries to address the challenge of limited experimental data by utilizing the rich information encapsulated in PLMs. This is a slightly different approach than the typical ways to RL or fine tuning to align the pLM to the experimental data.

• The paper introduces a dynamic biphasic annealing strategy for efficient sequence sampling, which maintains diversity while preventing premature convergence. The balance between cooling and heating phases is well-explained theoretically - and is shown to work in the insilico optimization experiments. It promotes exploration to obtain diverse high-fitness sequences. The strategy is sound, in particular with recent papers describing very rugged fitness landscapes observed in experimental dataset (Sandhu 2024).

• While the current work focuses on enzymes, the optimization algorithm seems reusable with any pLM, and any style of fine tuning, generally useful for any sequence fitness optimization problem.

Weaknesses – and suggestion for improvements

• The Seq2Fitness model is benchmarked against base ESM2. It is expected the semi supervised model will win – which is also shown clearly in later evaluations. A benchmark against typical techniques to adapt a pLM to experimental data would be good to get a fair comparison. Paper references (Hsu 2022, Nature) that has a list of good options, as does many other papers references. This could be classical fine tuning, LORA, or some reinforcement learning scheme like Preference Optimization. I would be very interested to seeing the difference.

• To show the semi supervised model works, another option is to run against a standard benchmark. The ProteinGym benchmark dataset (Notin, Neurips 2023) is a very commonly used benchmark, containing a collection of 200+ DMS experiments. Getting a comparative evaluation on this set would be very helpful, and would help compare directly with many more methods (including ESM2 zero shot).

• Experimental validation of BADASS: The paper uses 4 datasets for insilico fitness optimization as benchmarks (please described in more details e.g. in the suplementary material). Proper experimental validation invitro would make the paper much stronger. However, since not all groups have access to wet lab experimentation, it is an unfair ask. I can only suggest joining some of the optimization competitions to prove the worth of the method.

• As an alternative, although optimizing 4 backbones help focus on the individual datasets, it does not show the types of datasets that could be opttimized (potentially also antibodies, binders, etc). Testing on more -and more diverse- experimental backbones and different types of experimental assays would be useful to prove the generality of the method - but I would only encourage and it would not be a requirement.

• Limited benchmarking against alternative strategies: the method is benchmarked against EvoProdGrad and GSS. However, many other optimization approaches exist, that also attempts to optimize towards one or more virtual fitness scores. E.g Lambo2, Guided discrete diffusion (Gruver et al, 2023), or Discrete Walk Jump sampling (Frey et al, 2023). A comparison on the gain in virtual fitness against such competing methods would be interesting - or a method described in Hsu et al 2024.

Minor corrections

• Metrics: I believe the tables with metrics can be consensed significantly. The Top 100 metric and % Better than WT are not needed (perhaps just a line with the WT fitness on a figure?). Additional, typical criteria used of designability and e.g. structural metrics are not included - to show mutations do not reduce structural integrity (scored with Alphafold pLDDT).

• Please add a legend to Figure 9

Questions

• Could additional virtual fitness functions be introduced? Such as like AlphaFold pLDDT to assess foldability alongside fitness during the optimization process?

**Have the authors made all data and (if applicable) computational code underlying the findings in their manuscript fully available?**

Reviewer #1: Yes

Reviewer #2: Yes

Reviewer #3: None

PLOS authors have the option to publish the peer review history of their article (what does this mean?). If published, this will include your full peer review and any attached files.

Reviewer #1: No

Reviewer #2: No

Reviewer #3: No

**Figure resubmission:**
---

## [Decision Letter · Decision Letter 1]

PCOMPBIOL-D-24-01706R1

Designing diverse and high-performance proteins with a large language model in the loop

PLOS Computational Biology

Dear Dr. Gomez-Uribe,

Thank you for submitting your manuscript to PLOS Computational Biology. After careful consideration, we feel that it has merit but does not fully meet PLOS Computational Biology's publication criteria as it currently stands. Therefore, we invite you to submit a revised version of the manuscript that addresses the points raised during the review process.

Please submit your revised manuscript within 30 days (May 01, 2025; 11:59 PM). If you will need more time than this to complete your revisions, please reply to this message or contact the journal office at ploscompbiol@plos.org. Please include the following items when submitting your revised manuscript:

We look forward to receiving your revised manuscript.

Kind regards,

Mohammad Sadegh Taghizadeh, Ph.D.

Academic Editor

PLOS Computational Biology

Stacey Finley, Ph.D.

Section Editor

PLOS Computational Biology

**Journal Requirements:**

1) We noted that Figures (5-7) and Tables (4-17) are included in the Supplementary Information file. If they are supplemental ones,  please cite and label them as “S1 Table” and “S2 Table,” "S1 Figure", S2 Figure" and so forth. Please note that the supplementary figures should not be uploaded with the file type 'Figure'. Please amend the files type to 'Supporting Information'. Please ensure that each Supporting Information file has a legend listed in the manuscript after the references list.

2) Thank you for stating "The code and data have been available at the url indicated in our original submission for a few months now: https://github.com/SoluLearn/BADASS. Please update the Data Availability Statement in the online submission form to include the link of the dataset.

**Reviewers' comments:**

Reviewer's Responses to Questions

Reviewer #1: My comments were addressed.

Reviewer #2: The authors have sufficiently responded to the comments in their revision. They provide a novel protein engineering approach to design proteins with machine learning, integrating a semi-supervised neural network prediction model and an optimization algorithm to create novel amino acid sequences.

Reviewer #4: The authors present Seq2Fitness, a supervised deep learning model that integrates per-residue embeddings and log-probabilities from the ESM2 language model with a dual-path convolutional architecture and an MLP to predict protein variant fitness. The proposed method performs well on selected benchmarks and is further integrated into the authors' sequence design framework, BADASS. The overall idea is solid, and the combination of zero-shot signals with learned features is effective.

However, the following concerns should be addressed before the manuscript is considered for acceptance:

• The authors state: “For example, for both protein families 100% of the top 10,000 sequences found by BADASS have higher Seq2Fitness predictions than the wildtype sequence, versus less than 11% for competing approaches.”

Both BADASS and Seq2Fitness are methods developed by the authors and are tightly integrated for directed evolution. Since BADASS directly depends on Seq2Fitness for scoring, comparing its output to sequences generated by competing approaches—especially when evaluation is based on Seq2Fitness predictions—is potentially biased and not a fair comparison. This limitation should be acknowledged.

• The model is trained and evaluated on datasets corresponding to few proteins. As a result, Seq2Fitness is not currently generalizable across proteins. This important limitation should be clearly stated wherever the performance or applicability of the algorithm is described.

• It would be helpful if the authors could comment on the potential of training Seq2Fitness on larger and more diverse datasets, such as ProTherm, which contains mutation data across a wide range of human proteins, including both single and multiple mutations. This could significantly improve the model’s generalizability.

• The manuscript includes the following sentence: “ESM2 Model: a zero-shot model for scoring protein sequences based on their unsupervised representations. We use the ESM2 650M model, and the mutant marginal score as the fitness. (add equation here?)”

Remove the phrase “(add equation here?)”

• The authors note: “We evaluated designed sequences containing 2–6 mutations relative to a reference sequence.”

Is the reference sequence periodically updated? If not, it would be useful to explain why the reference sequence was not iteratively updated during sequence generation. Iterative refinement of the reference is a standard approach in directed evolution and could improve sequence exploration.

• Finally, the evaluation of Seq2Fitness is based on an 80/20 train-test split. It would be more appropriate to use 5-fold cross-validation to obtain a more robust estimate of model performance, especially given the relatively small dataset sizes. While leave-two-out cross-validation may be computationally expensive, 5-fold cross-validation is a reasonable and standard compromise.

**Have the authors made all data and (if applicable) computational code underlying the findings in their manuscript fully available?**

Reviewer #1: Yes

Reviewer #2: Yes

Reviewer #4: Yes

PLOS authors have the option to publish the peer review history of their article (what does this mean?). If published, this will include your full peer review and any attached files.

Reviewer #1: **Yes: **Yi Xiong

Reviewer #2: No

Reviewer #4: No

**Figure resubmission:**
---

## [Decision Letter · Decision Letter 2]

Dear Dr. Gomez-Uribe,

We are pleased to inform you that your manuscript 'Designing diverse and high-performance proteins with a large language model in the loop' has been provisionally accepted for publication in PLOS Computational Biology.

Best regards,

Mohammad Sadegh Taghizadeh, Ph.D.

Academic Editor

PLOS Computational Biology

Stacey Finley, Ph.D.

Section Editor

PLOS Computational Biology

Reviewer's Responses to Questions

**Comments to the Authors:**

Reviewer #4: The authors have addressed my concerns in a satisfactory way. The manuscript can now be accepted.

**Have the authors made all data and (if applicable) computational code underlying the findings in their manuscript fully available?**

Reviewer #4: Yes

PLOS authors have the option to publish the peer review history of their article (what does this mean?). If published, this will include your full peer review and any attached files.

Reviewer #4: No

---

## [Editor Report · Acceptance letter]

PCOMPBIOL-D-24-01706R2

Designing diverse and high-performance proteins with a large

language model in the loop

Dear Dr Gomez-Uribe,

I am pleased to inform you that your manuscript has been formally accepted for publication in PLOS Computational Biology. Your manuscript is now with our production department and you will be notified of the publication date in due course.

With kind regards,

Anita Estes
